# Double-layered protein nanoparticles induce broad protection against divergent influenza A viruses

Lei Deng[1], Teena Mohan[1], Timothy Z. Chang[2], Gilbert X. Gonzalez[1], Ye Wang[1], Young-Man Kwon[1], Sang-Moo Kang[1], Richard W. Compans[3], Julie A. Champion[2] & Bao-Zhong Wang [1]

Current influenza vaccines provide limited protection against circulating influenza A viruses. A universal influenza vaccine will eliminate the intrinsic limitations of the seasonal flu vaccines. Here we report methodology to generate double-layered protein nanoparticles as a universal influenza vaccine. Layered nanoparticles are fabricated by desolvating tetrameric M2e into protein nanoparticle cores and coating these cores by crosslinking headless HAs. Representative headless HAs of two HA phylogenetic groups are constructed and purified. Vaccinations with the resulting protein nanoparticles in mice induces robust long-lasting immunity, fully protecting the mice against challenges by divergent influenza A viruses of the same group or both groups. The results demonstrate the importance of incorporating both structure-stabilized HA stalk domains and M2e into a universal influenza vaccine to improve its protective potency and breadth. These potent disassemblable protein nanoparticles indicate a wide application in protein drug delivery and controlled release.

[1] Center for Inflammation, Immunity & Infection, Georgia State University, Atlanta, GA 30303, USA. [2] School of Chemical & Biomolecular Engineering, Georgia Institute of Technology, Atlanta, GA 30332, USA. [3] Department of Microbiology and Immunology and Emory Vaccine Center, Emory University School of Medicine, Atlanta, GA 30322, USA. Correspondence and requests for materials should be addressed to B.-Z.W. (email: bwang23@gsu.edu)

**M**utant viruses acquire the ability to escape from prevailing herd immunity by antigenic drift and shift, which necessitates the yearly update of the composition of seasonal influenza vaccines to match the newly circulating viruses[1]. The protective efficacy of the seasonal vaccines does not always live up to expectation. The outbreak of 2009 H1N1 pandemic caused 200,000 deaths during the first 12 months of its circulation[2]. Low vaccine effectiveness was also observed recently during the 2012–2013 and 2014–2015 flu seasons[3,4]. The sporadic human cases of fatal zoonotic H5N1 and H7N9 infections are also serious public health threats[5–7]. A universal influenza vaccine which induces broad cross protection against divergent viruses is urgently needed to eliminate these threats. Conserved determinants from influenza antigenic proteins are potential immunogens for such universal influenza vaccines. The HA stalk domain is relatively conserved compared to the variable globular head domain[8,9]. Accompanying the isolation and artificial generation of broadly neutralizing antibodies[10–16], some HA stalk domain-based immunogens have been constructed and proven protective to some extent in vivo[17–19].

The amino acid sequence of influenza matrix protein 2 ectodomain (M2e) is highly conserved among human seasonal influenza A viruses[20]. Natural human influenza A virus infections induce only weak anti-M2e antibody responses of short duration[21]. A possible explanation for this low immunogenicity is the small size of M2e and the low abundance of M2 in virions compared to the large glycoproteins, HA, and NA[22]. Therefore, M2e is often constructed with a larger carrier or presented as a soluble tetramer antigen to enhance anti-M2e immune responses in vaccination experiments[23,24]. Multiple copies of M2e in a construct can dramatically enhance the M2e specific antibody responses[25]. Clinical trials have demonstrated that M2e based vaccines are safe and immunogenic in humans[20,26,27]. Human passive immunization with humanized anti-M2e monoclonal antibody TCN-032 proved to be efficient in reducing virus replication, demonstrating the effectiveness of vaccine-driven anti-M2e antibody-based immunity[28]. Clinical trial results have shown that the overall induced M2e antibody responses in M2e-HBc vaccinated volunteers faded away rapidly within 10 months[20].

Successful applications of nanotechnology hold great promise for the development of new generations of influenza vaccines. Large self-assembling motifs can enable 24-mer[17,29] and even 60-mer[30] protein nanoparticle (PNp) assembly. However, self-assembly motifs increase the risk of off-target immune responses due to their high immunogenicity. Desolvated PNp core coated with viral antigen on the surface represents a convenient solution to these issues and does not require encapsulation materials.

In this study, we found that layered PNps composed of structure-stabilized HA stalk domains from both HA groups, and novel constructed M2e, are highly immunogenic to induce immune protection against homosubtypic and heterosubtypic influenza A virus challenges. The double-layered PNps have the potentials to be developed into a universal influenza vaccine. The physiologically activated disassembly of PNps after the uptake into cells implies a wide utilization for protein drug delivery and controlled release.

## Results

**Characterization of recombinant proteins and nanoparticles.** We successfully constructed and expressed the structure-

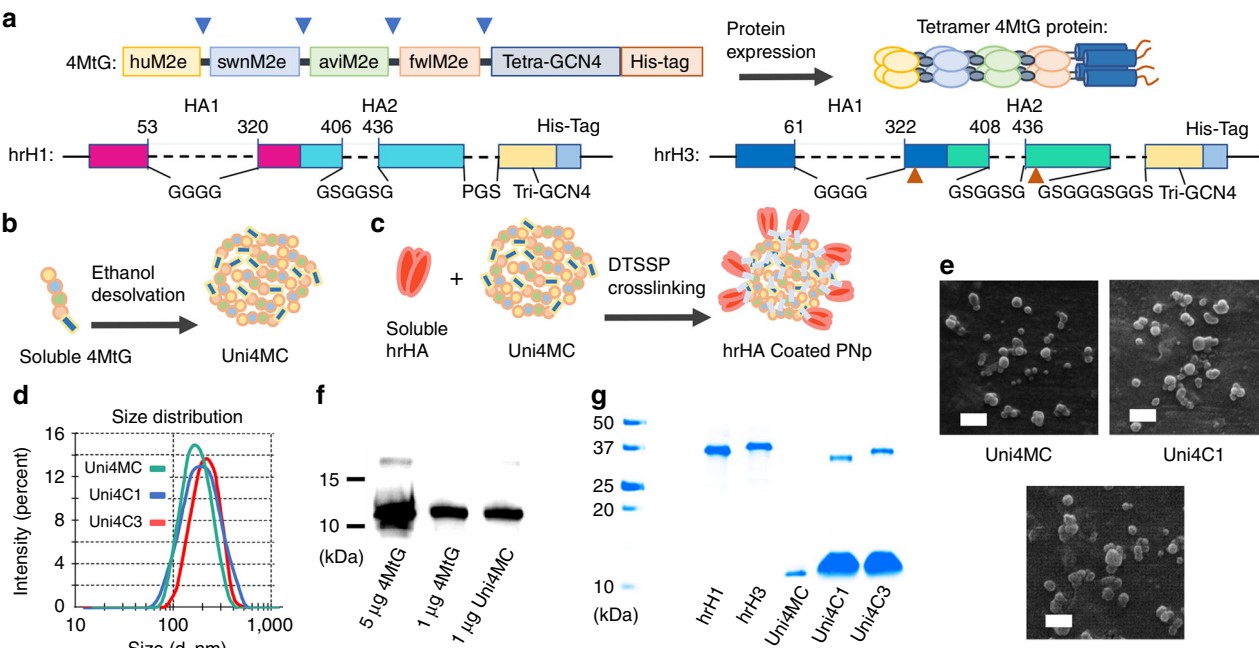

**Fig. 1** Recombinant protein construction and PNp generation and characterization. **a** Cartoon models of the construction and expression of recombinant proteins 4MtG, hrH1, and hrH3. The numbering of hrH1 and hrH3 are based on the amino acid sequences of PR8 and Aic HA, respectively. Blue arrows indicate the location of flexible linkers. Dashed lines indicate the sequences replaced with linkers shown below. Brown arrows indicate the site-mutations, V325C and S438C. **b** Schematic diagram of Uni4MC (desolvated 4MtG PNp) fabrication. Recombinant 4MtG protein was self-assembled into PNps by desolvation as described in the Materials and Methods. **c** Schematic diagram of double-layered nanoparticle generation. An additional layer of trimeric hrHA proteins was crosslinked onto the desolvated Uni4MC PNp surface via DTSSP crosslinking. **d** Size distribution of Uni4MC, Uni4C1 (hrH1-coated double-layer PNp) and Uni4C3 (hrH3-coated double-layer PNp). **e** Scanning electron microscopy of Uni4MC, Uni4C1, and Uni4C3 PNps (Bars represent 500 nm in length). **f** Western blotting analysis of soluble 4MtG protein and Uni4MC PNps. **g** Coomassie blue staining analysis of soluble proteins hrH1 and hrH3, Uni4MC PNp, and double-layer Uni4C1 and Uni4C3 PNps

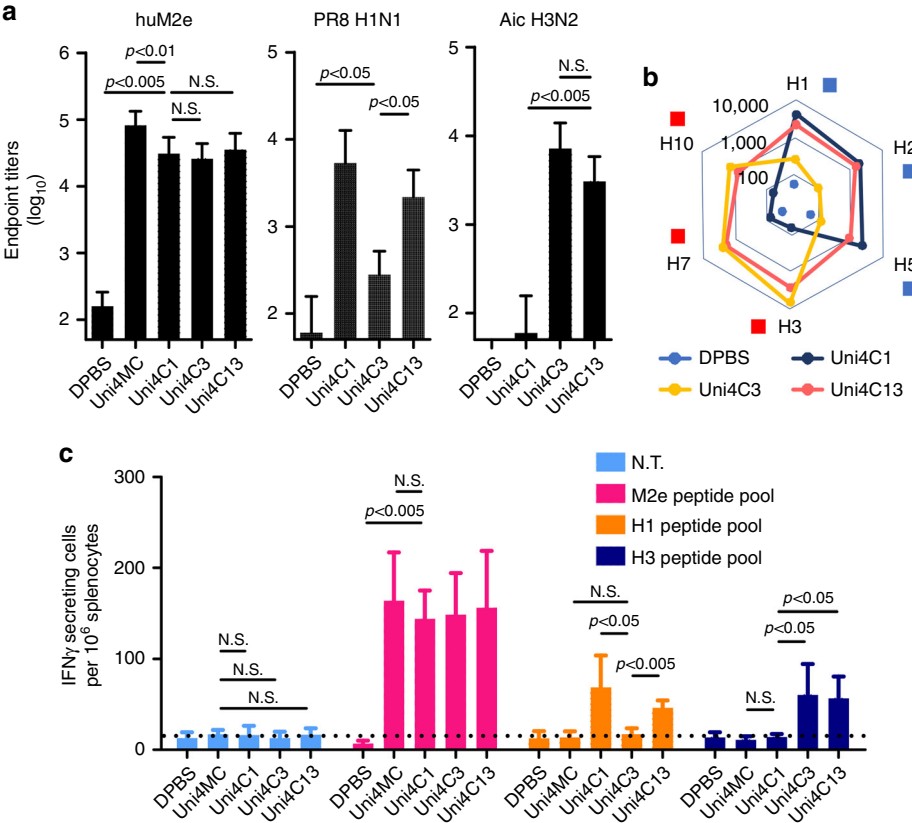

**Fig. 2** Humoral and cellular immune responses of vaccinated mice. **a** Serum IgG endpoint titers against huM2e, formalin-inactivated PR8 H1N1 or formalin-inactivated Aic H3N2. ($n = 10$) **b** Radar diagram depicting breadth of immune serum binding to HA subtypes. Because frequent human infection with fatal zoonotic influenza H5N1, H7N9, and H10N8 viruses in recent years represents possible emerging pandemics, the binding activities of immune sera to H5, H7, and H10 were tested. The waning human herd immunity against H2N2 influenza strain and low mutation rate make it likely that a new pandemic could arise from the current circulating H2N2 strain among birds and swine. H2 was included in the serum binding assay. The squares besides HA subtypes indicate HA phylogenetic groups by coloring: blue for group 1 and orange for group 2. **c** Specific cellular immune responses against M2e, H1, and H3. Peptide pool re-stimulated interferon gamma (IFNγ)-secreting cell clones were determined using ELISpot assay. ($n = 4$) Data are presented as mean ± SD Statistical significance was analyzed by t-test for **a** and **c**. P values shown in bar charts and N.S. indicates no significance between two compared groups. The experiments were repeated twice with similar results

stabilized soluble tetramer protein containing four tandem copies of M2e (4MtG) and trimeric head-removed (hr) HAs from representatives of both HA phylogenetic groups (designed hrH1 and hrH3). To increase the potency and breadth of protection, 4MtG includes four types of M2e from human, swine, avian, and domestic fowl viral consensus sequences (Fig. 1a, upper; sequences in Supplementary Table 1). The generation of hrHA representatives from both phylogenetic groups is essential to formulate a vaccine that can elicit a genuinely universal protective response.

Starting with the HA sequences of A/Puerto Rico/8/1934 (H1N1, PR8) from group 1 and A/Aichi/2/1968 (H3N2, Aic) from group 2, a series of structure-based mutations was introduced. Due to its metastable conformation, HA2 expressed independently from the HA1 subunit will spontaneously adopt a low-pH conformation[31]. Here, hrH1 and hrH3 antigens are composed of the stalk region of HA1 and the ectodomain of HA2 (Fig. 1a, bottom; Supplementary Fig. 1a, b). The polymerization of 4MtG and hrHAs was demonstrated by bis [sulfosuccinimidyl] (BS3) crosslinking followed by SDS-PAGE Coomassie Blue staining or Western blots (Supplementary Fig. 2a–c). Strong binding of 4MtG to mAb 14C2 indicated the antigenicity of the purified 4MtG protein (Supplementary Fig. 2d). The antigenicity of hrH1 and hrH3 was evidenced by the strong binding of hrH1 to conformation-specific mAb C179 and hrH3 to linear-epitope-

recognizing mAb 12D1 and conformation-specific mAb 9H10 in sandwich ELISA (Supplementary Fig. 2e–g). This indicates that the recombinant hrH1 and hrH3 proteins were folded in a similar fashion to the corresponding region in HA in the neutral pH structure.

We generated double-layered PNps by desolvation and 3,3′-dithiobis (sulfosuccinimidyl propionate) (DTSSP) crosslinking[32–34]. This approach produced PNps composed almost entirely of the antigens of interest and did not include carriers, avoiding off-target immune responses to vectors[17,29,35]. Desolvated M2e PNps proved remarkably immunogenic[34], and coated double-layer HA PNps better retained HA hemagglutination function[32]. Thus, we desolvated 4MtG into PNps (designated Uni4MC) as the core as diagrammed in Fig. 1b and crosslinked hrHA as a coating layer to resemble virion size and surface antigen display, as diagrammed in Fig. 1c (hrH1-coated double-layer PNps designated Uni4C1; and hrH3-coated PNps designated Uni4C3).

PNp size expanded with increasing amounts of DTSSP used during desolvation (Supplementary Fig. 3a, b) but was weakly influenced by the desolvation ratio (Supplementary Fig. 3c, d). We have shown that PNps with small sizes induce significant upregulation of the pro-inflammatory cytokine IL-1β from dendritic cells compared to soluble antigens, potentially enhancing specific immune responses[33].

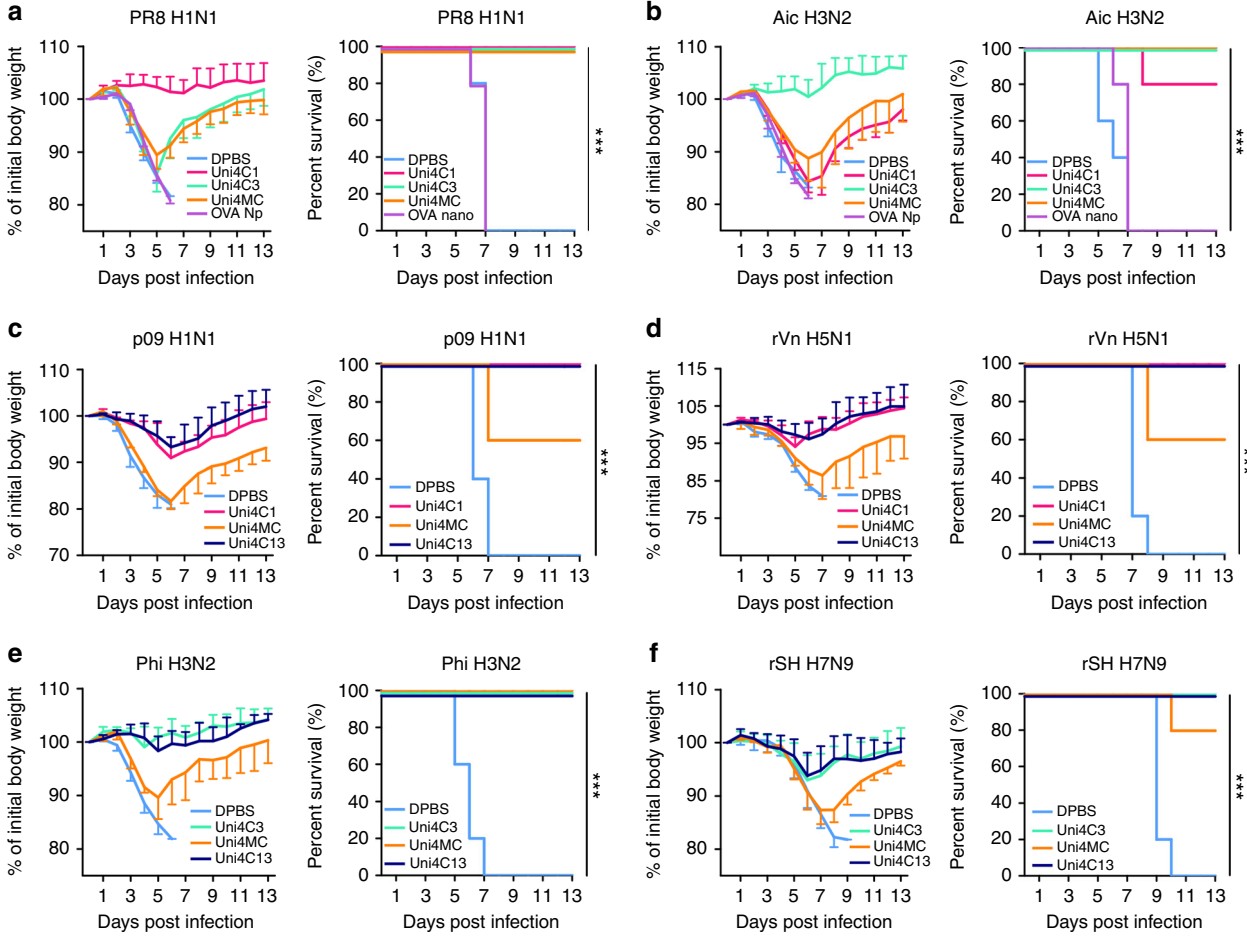

**Fig. 3** PNp protective efficacy in mice. Mean body weight changes and survivals upon lethal dose challenges. Error bars represent mean ± SD and statistical analysis was performed with a log-rank test (***$p < 0.001$, $n = 5$). For viral challenges, mice were slightly anesthetized by inhalation of isoflurane, and a dose of 6 mLD$_{50}$ virus in 50 µl PBS was dropped into two nares. Mouse body weight changes and survivals were monitored for 2 weeks. **a** PR8; **b** Aic; **c** p09; **d** reassortant Vtn (rVn); **e** Phi; and **f** reassortant SH (rSH)

To minimize the PNp size, the core was not crosslinked until after the addition of soluble hrHA during the coating process because proteins self-assembled into smaller PNps in the absence of a crosslinker (Supplementary Fig. 3e). Dynamic light scattering (DLS) analysis showed an average hydrodynamic diameter of 178 nm for Uni4MC, 194.8 nm for Uni4C1, and 228.5 nm for Uni4C3 (Fig. 1d). Scanning electron micrograph (SEM) showed that the particles were relatively spherical with irregular surface morphology (Fig. 1e). Quantification analysis (by using ImageQuantTL software) of the PNps sizes showed the range of distribution of Uni4MC from 80 nm to 260 nm with 96% PNp in the range of 120–240 nm, Uni4C1 from 120 nm to 300+ nm with 87% PNp in the range of 160– 240 nm, and Uni4C3 from 80 nm to 300+ nm, with 85% PNp in the range of 120–240 nm (Supplementary Fig. 3f). There was no significant difference among the three nanoparticle types (Supplementary Fig. 3g).

Uni4MC can be completely disassembled and reduced into monomers, and blotted by anti-M2e serum in Western Blot (Fig. 1f). Intensity analysis of the protein bands in Coomassie Blue stained gel estimated the overall percentages of hrH1 and hrH3 coatings to be 11.5 and 14% in the resultant double-layer PNps, respectively (Fig. 1g).

The unbound hrHA molecules and self-crosslinked hrHAs cannot be pelleted by high-speed centrifugation[36]. In the pull-down assay, we found that no uncoated 4MtG PNps existed in Uni4C1 and Uni4C3 samples (Supplementary Fig. 3h). Immuno-

gold labelling and transmission electronic microscopy imaging showed C179-stained gold spotted Uni4C1, 12D1-stained gold-spotted Uni4C3 and unspotted Uni4MC (Supplementary Fig. 3i), demonstrating the coating of hrHAs onto particle surfaces.

**PNps induce robust immunity conferring complete protection.** To evaluate the protective efficacy of the resulting PNp vaccines, mice were intramuscularly immunized twice with PNps in the absence of adjuvants. Strong seroconversion against M2e, H1N1 (PR8), and H3N2 (Aic) was elicited after boosting immunizations (Fig. 2a; Supplementary Fig. 4a). The induced M2e antibodies showed strong cross-reactivity to diverse M2e peptides including A/California/7/2009 (H1N1, p09) M2e, A/Vietnam/1203/2004 (H5N1, Vtn) M2e and A/Shanghai/2/2013 (H7N9, SH) M2e (Supplementary Table 1 and Supplementary Fig. 4b). Both Uni4C1- and Uni4C3-elicited sera bound strongly to HA homologous to the vaccine strain but weakly to heterologous HA (Fig. 2a, b). Of note, Uni4C13 (cocktail of Uni4C1 and Uni4C3) elicited sera strongly and broadly reactive to various HA antigens, including inactivated PR8, Aic, reassortant Vtn (rVn), or reassortant SH (rSH), or full length (FL) HAs from J57 H2N2 (strain: A/Japan/305/1957) or GD H10N8 (strain: A/duck/Guangdong/E1/2012) expressed on HEK293T cell surfaces in cell surface ELISA (Fig. 2b; Supplementary Fig. 4c). Antibody isotypes against M2e, inactivated PR8 and Aic were evaluated (Supplementary

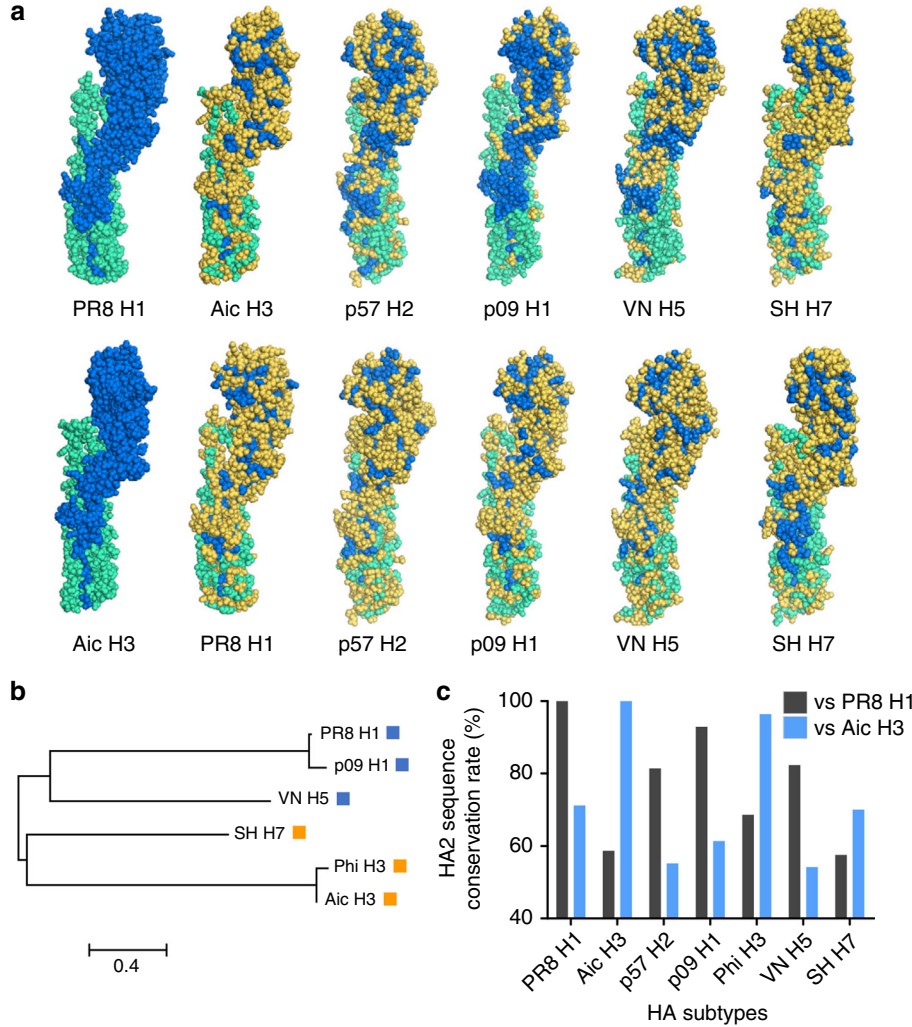

**Fig. 4** Amino acid differences in HA structures. **a** Three-dimensional spherical structures of diverse hemagglutinin proteins with amino acid variations against PR8 H1 or Aic H3. The HA1 domain is shown in green-cyan and the HA2 domain is shown in blue. The different residues in other HA proteins against PR8 H1 or Aic H3 are colored in yellow. (PR8 H1 PDB: 1rvx; Aic H3 PDB: 3ztj; p57 H2 PDB: 4hf5; p09 H1 PDB: 5gjs; VN H5 PDB: 2fk0; SH H7 PDB: 5v2a). **b** Phylogenetic tree of HAs. The squares besides HA subtypes indicate HA phylogenetic groups by coloring: blue for group 1 and orange for group 2. **c** HA2 domain sequence conservation rate against PR8 H1 and Aic H3

Fig. 4d). IgG2a-biased antibody responses were induced, as evidenced by the IgG1:IgG2a ratio (Supplementary Fig. 4e).

Cellular responses are also important for the generation and regulation of effective immunity and contribute to the killing of pathogens[37,38]. As shown in Fig. 2c, the Uni4C13 group demonstrated broad cellular responses, and induced significantly more IFN-γ secreting splenocyte populations after re-stimulation with diverse antigen peptide pools.

Prophylaxis potency was evaluated by challenge studies. Four weeks after the boosting immunization, $6 \times 50\%$ mouse lethal doses (mLD$_{50}$) of mouse-adapted influenza A viruses were used for intranasal challenges. Uni4C1 and Uni4C3 immunizations conferred complete homologous protection against death and weight loss (Fig. 3a, b), whereas Uni4MC-immunized mice suffered severe weight loss. The OVA PNp-immunized mice did not survive challenges and experienced a similar weight drop to the Dulbecco's phosphate-buffered saline (DPBS) group, excluding the possible role of non-specific particulate motifs in providing protection. Immunizations with Uni4C13 but neither Uni4C1 nor Uni4C3 provided universally complete protection against lethal viral challenges (Fig. 3c–f). Both challenge strains reassortant H5N1 (rVn) and H7N9 (rSH) bear internal genes

from PR8 H1N1, including the same PR8 M2e sequence (Supplementary Table 1). The conservation between divergent subtypes of HA is quite limited (Fig. 4a–c). Moreover, the Uni4C13 immunization dramatically reduced lung virus titers at day 5 post infection compared to the mock-immunization (Fig. 5a, b). Histological analysis showed significantly decreased leukocyte infiltration in infected lungs from Uni4C13 immunized mice (Fig. 5c, d), and the score for leukocyte infiltration in this group was significantly lower than other groups (Fig. 5e, f).

**Antibodies correlate with immune protection.** We next investigated the antibody-mediated effector mechanisms. No hemagglutination-inhibition (HAI) activities were detected with pre-challenge sera (Table 1). Measurement of serum neutralization (NT) revealed appreciable NT activity in Uni4C1 and Uni4C3 immune sera against homologous strains. Uni4C13 immune sera showed NT activity against both H1, H3 and H5 subtypes (Table 2). NT activity against rSH was undetectable.

Besides direct viral NT, Fc-mediated effector mechanisms such as antibody dependent cellular cytotoxicity (ADCC) contribute substantially to protection against influenza challenges[10,39]. In

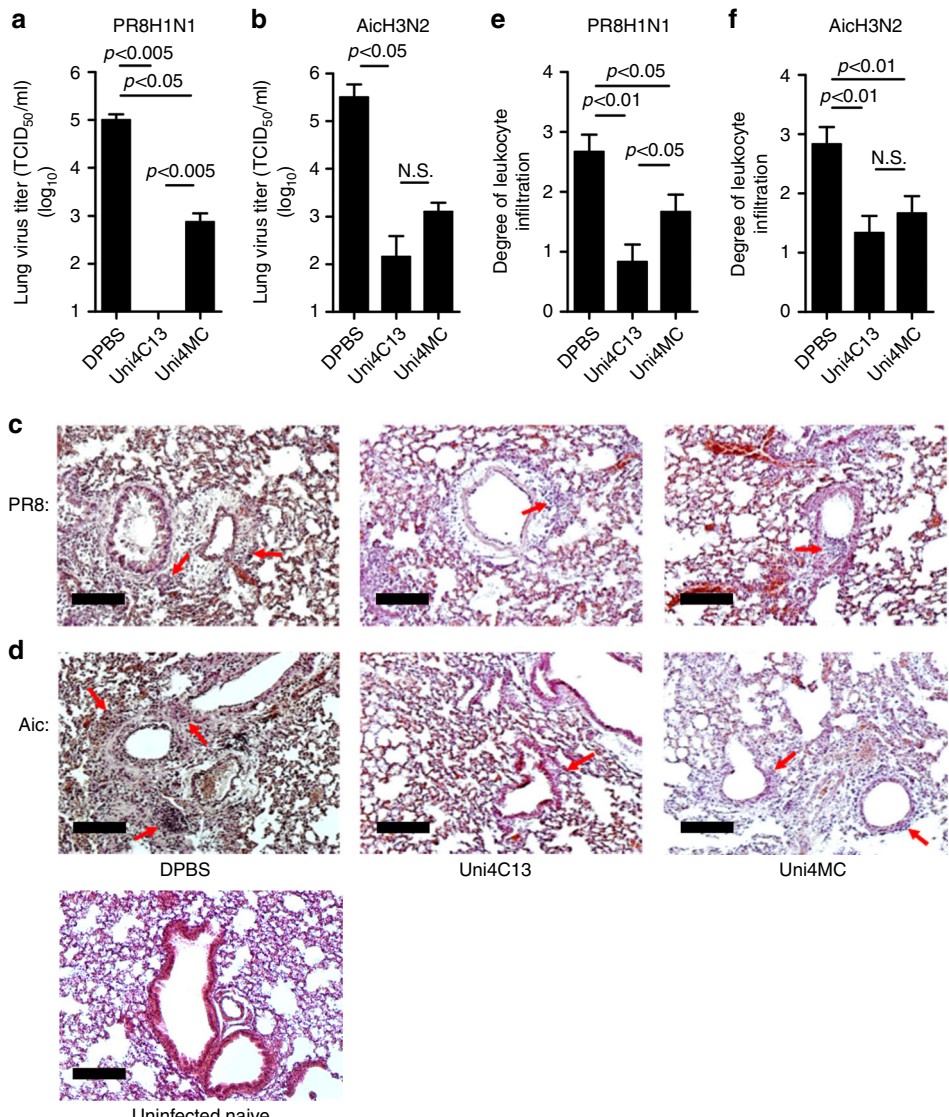

**Fig. 5** Lung physiology in virus challenge infection. **a**, **b** Determination of mouse lung virus titers at day 5 post a sublethal dose infection with PR8 H1N1 (**a**) or Aic H3N2 (**b**). **c**, **d** Histological pathology analysis. Red arrows in images indicate leukocyte infiltration in lung sections from mice infected with PR8 H1N1 (**c**) or Aic H3N2 (**d**). Uninfected lung section is used as negative control. (Bars represent 400 μm in length) **e**, **f** Bar charts showing the scores of leukocyte infiltrations degree after infection with PR8 H1N1 (**e**) or Aic H3N2 (**f**). Data are presented as mean ± SD ($n = 5$) and statistical significance was analyzed by t-test for **a**, **b**, **e** and **f**. P values shown in bar charts and N.S. indicates no significance between two compared groups. The experiments were repeated twice with similar results. Images from **c** and **d** are representatives from five individual mouse lung sections each group

accordance with the binding data, Uni4C13 induced serum antibodies with wide ranging ADCC potency against target cells expressing M2, FL H1 or FL H3 (Fig. 6a–c). We passively transferred immune sera from mice immunized with DPBS, Uni4MC or Uni4C13 to three groups of naïve mice ($n = 5$). After 24 h, mice were intranasally challenged with $3 \times mLD_{50}$ of PR8 or Aic (Fig. 6d, e). In contrast to mice that received mock-sera, mice that received sera from Uni4C13-immunized mice were protected significantly, indicating that serum antibodies played critical roles in the protection conferred by Uni4C13.

Clodronate-loaded liposomes were intra-tracheally administered for selective depletion of alveolar macrophages (AM). Passively transferred Uni4C13 serum in both PBS-loaded and clodronate-loaded liposome treated mice have similar antibody titers as evidenced in M2e peptide based ELISA (Fig. 6f). Challenge results showed that only mice receiving Uni4C13 serum and without the AM depletion were fully protected against lethal dose Phi H3N2 challenge, indicating an

antibody-dependent cellular phagocytosis (ADCP) mechanism (Fig. 6g). An antibody titer follow-up study shows that PNp-induced antibodies specific to human M2e, PR8, and Aic were durable up to 4 months after the last immunization in mice (Fig. 7), implicating a long-lasting protection.

## Discussion

An influenza vaccine which induces broad cross protection against various influenza viruses has been a scientific challenge for more than a half century[40,41]. With the recognition that immunity specific to conserved epitopes in influenza antigenic proteins is the "holy grail" of such protection, increasing evidence indicates that the conserved HA stalk region and M2e have the potentials to be developed into a universal influenza vaccine, if these epitopes can be appropriately presented and sensed by the host immune system[35,42,43]. A promising approach is to express a standalone HA stalk domain without the highly immunogenic

**Table 1 HAI titers of immune sera prior to and post challenges**

| Virus subtypes | Pre-challenge sera three weeks post boost | | | | | | Subunit protein sera or convalescent sera | | | |
| --- | --- | --- | --- | --- | --- | --- | --- | --- | --- | --- |
| | Pre-immune | DPBS | Uni4MC | Uni4C1 | Uni4C3 | Uni4C13 | PR8 H1 | Aic H3 | rVn H5 | rSH H7 |
| PR8 H1N1 | <10 | <10 | <10 | <10 | <10 | <10 | 160 | <10 | 40 | <10 |
| Aic H3N2 | <10 | <10 | <10 | <10 | <10 | <10 | <10 | 640 | <10 | <10 |
| rVn H5N1 | <10 | <10 | <10 | <10 | <10 | <10 | <10 | <10 | 160 | <10 |
| rSH H7N9 | <10 | <10 | <10 | <10 | <10 | <10 | <10 | <10 | <10 | 80 |
| p68 H3N2 | <10 | <10 | <10 | <10 | <10 | <10 | <10 | 640 | <10 | <10 |

Low titer ▬▬▬ High titer

A panel of viruses including PR8, Aic, rVn, rSH, and A/Hong Kong/1/1968 (H3N2, p68) were tested for HAI. Immune sera against HA proteins PR8 H1, Aic H3, and rSH H7 and convalescent sera from rVn virus infection were used as positive controls. The lowest serum dilution able to inhibit virus hemagglutination is shown ($n = 10$)

**Table 2 Virus neutralization titers**

| Serum samples | Virus subtypes | | | | |
| --- | --- | --- | --- | --- | --- |
| | PR8 H1N1 | Aic H3N2 | p68 H3N2 | rVn H5N1 | rSH H7N9 |
| DPBS | <10 | <10 | <10 | <10 | <10 |
| Uni4MC | <10 | <10 | <10 | <10 | <10 |
| Uni4C1 | 80 | <10 | <10 | 20 | <10 |
| Uni4C3 | <10 | 320 | 320 | <10 | <10 |
| Uni4C13 | 80 | 320 | 320 | 20 | <10 |
| PR8 H1 immune serum | 640 | <10 | <10 | <10 | <10 |
| Aic H3 immune serum | <10 | >1280 | >1280 | <10 | <10 |
| rVn H5N1 convalescent | 10 | <10 | <10 | 80 | <10 |
| rSH H7 immune serum | <10 | <10 | <10 | <10 | 80 |

Low titer ▬▬▬ High titer

Virus neutralization potency of immune sera was evaluated in a standard neutralization assay against the same panel of influenza viruses as in the HAI assays. The lowest serum dilution able to neutralize $100 \times TCID_{50}$ of a given virus is shown ($n = 10$)

and strain-specific head domain, and a stabilized M2e with high epitope density. It was reported that HA stalk sera strongly bind HA from the same phylogenetic group but not the heterosubtypic HA from the different groups. Increasing evidence also showed that influenza genome of human isolates contained M2 gene segment from swine and avian origins, like p09 H1N1, human H5N1, and H7N9. Therefore, we started generating HA stalk constructs from PR8 H1 (group 1) and Aic H3 (group 2), as well as the M2e construct expressing tandem copies of M2e from human, swine, avian, and domestic fowl influenza.

Our study provides insights into a novel format of particulate influenza vaccines, which contain only antigenic proteins of interest and can induce robust and long-lasting immunity. We fabricated PNps approximately the size of the influenza A virions with a core of M2e displaying a shell of conserved hrHA domains. The binding of soluble hrHA to the desolvated PNps is speculated to be mediated through interaction of hydrophobic residues and fixed by DTSSP crosslinking primary amines. To our knowledge, this design avoids the risk of instability shown by virus-like particles under osmotic stresses or during changes in salt concentration and prevents off-target immune responses against self-assembly motifs, such as the ferritin[17,29] or hepatitis B core[35] protein used in some PNp designs.

DTSSP is a water-soluble, thiol-cleavable and primary amine-reactive cross-linker[44]. The crosslinked PNp with higher density than water can be pelleted after high speed centrifugation. Due to the reducibility of DTSSP, the cross-linker can be compromised by the abundance of intracellular thiols, thus PNps can slowly release free protein molecules after uptake by APCs. Enhancing redox responsiveness has been shown to be effective in boosting the immune response to reduced antigens[45]. The physiological activated release of antigenic proteins from PNps may significantly contribute to the broad immune protection seen.

Immunization with Uni4C13 PNp induced remarkably high level of hrHA-specific and M2e-specific immune responses. It was found that nanoparticulate antigens are more efficiently taken up by phagocytes than soluble antigens, promoting dendritic cell (DC) maturation and stimulating IL-1β production in bone marrow derived DC[32]. Less actin polymerization was required in PNp uptake by DC than soluble antigen. It was reported that sustained antigen releasing shapes memory T cells and is responsible for long-term immunity[46]. However, the immuno-modulatory effects of PNp have been shown to be a function of many aspects, including size[47,48], shape[49,50], charge[51], surface chemistry[52] and administration route[32,34]. There is no universal particulate design principle yet.

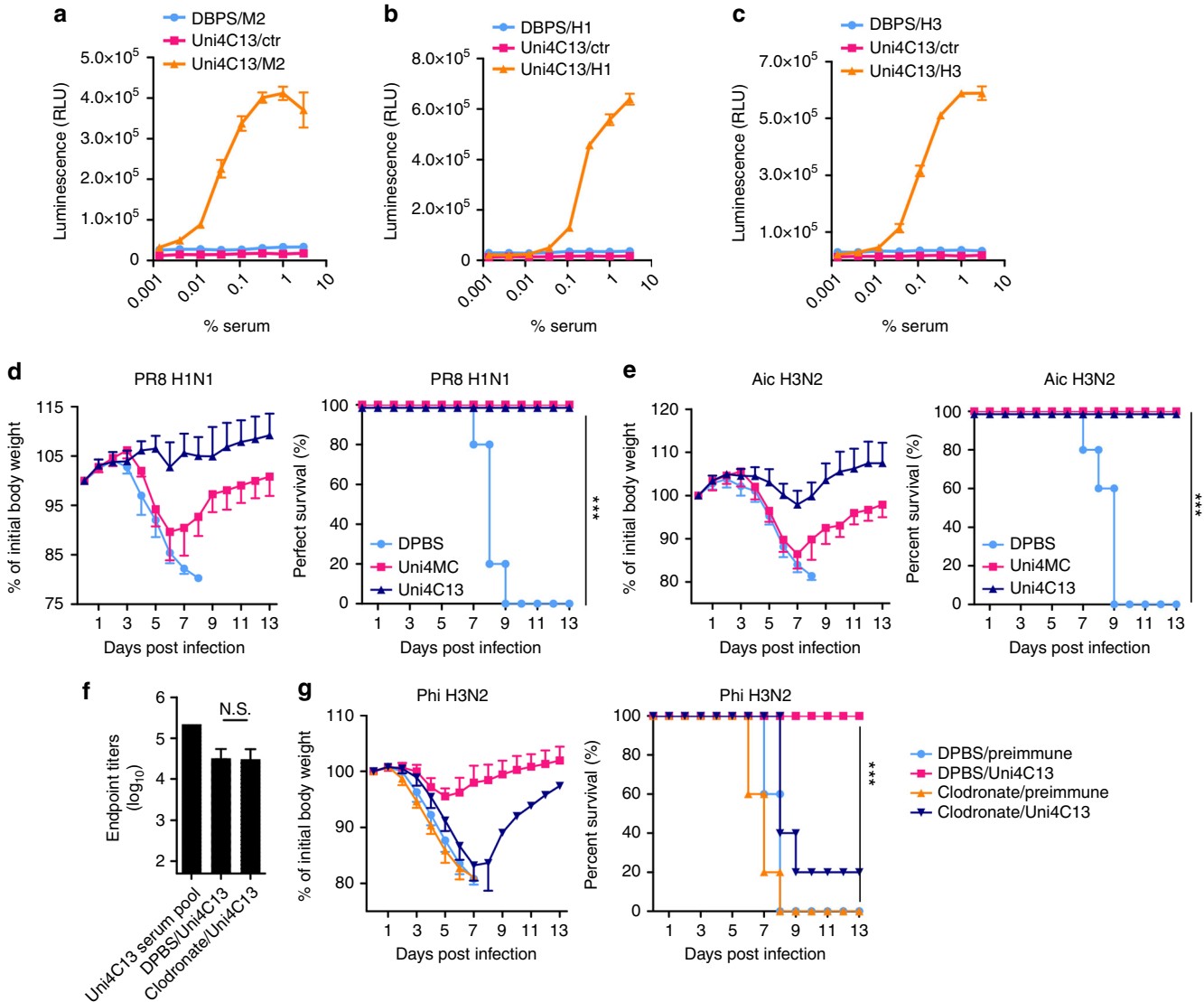

**Fig. 6** Correlates of immune Protection. **a–c** ADCC surrogate assay with pooled pre-challenge sera from each immunization group and transiently transfected HEK293T target cells expressing M2 (**a**), H1(**b**) or H3 (**c**). The non-transfected HEK293T cells were used as negative control. ($n = 3$) RLU: Relative light units. Data are presented as mean ± SD. The experiments were repeated twice with similar results. **d–g** Protection by passive transfer of immune sera. Immune sera from mice immunized with Uni4MC, Uni4C13 or mock, respectively, were pooled and transferred to naïve recipient mice 24 h before challenge with $3 \times mLD_{50}$ PR8 H1N1 (**d**) or Aic H3N2 (**e**). **f** Left bar represents the anti-M2e IgG endpoint titer in Uni4C13 serum pool that was used for intraperitoneal passive transfer. Other two bars show the anti-M2e IgG endpoint titers in PBS loaded or clodronate-loaded liposomes treated recipient mice. **g** Morbidity and mortality of BALB/c mice receiving pre-immune or Uni4C13 serum, depleted or not depleted of alveolar macrophages, then were challenged with $3 \times mLD_{50}$ Phi H3N2. Body weight loss and survival rate were monitored for 14 days. Statistical significance analysis of survival rates was performed by Kaplan-Meier analysis for **d–g**. ***$p < 0.005$. Data are presented as mean ± SD ($n = 5$) and statistical significance was analyzed by $t$-test for **f**. N.S. indicates no significance between two compared groups. Data are representative from two independent experiments

We have shown that PNp vaccines based on both M2e and HA stalk domains protect BALB/c mice mainly through ADCC and ADCP. This result is consistent with previous findings on the M2e-based and HA stalk-based immunity mechanism[18,53–56]. BALB/c mouse IgG2a is more efficient than IgG1 in Fc-mediated effector functions, capable of interacting with all activating FcγRI, FcγRIII, and FcγRIV[57]. Our data indicate an IgG2a-biased antibody response after M2e PNp vaccination and IgG1-biased antibody response after H7 PNp immunization[32,34].

In conclusion, the structure-based designs of hrHA and 4MtG recombinant proteins and the novel PNp structures provide proof-of-concept for a universal influenza vaccine. Moreover, the potent disassemblable PNps indicate a wide application for the development of vaccines against other pathogens, and for protein

drug delivery and controlled release. Coating PNps with immune cell receptor ligands such as CD154, complement receptor 1/2 ligands and so on, or with protein adjuvants such as fusokines, TLR ligands and C–C motif chemokine ligands, is another avenue to facilitate targeted protein drug delivery and to enhance immune responses. In addition, the abiotic nature of PNps also enhances their amenability to cold chain-independent storage for 3 months at room temperature, a desirable property for vaccine transport to low-income countries.

## Methods

**Ethics statement**. This study was carried out in strict accordance with the recommendations in the Guide for the Care and Use of Laboratory Animals of the National Institutes of Health. All mouse studies were approved by Georgia State

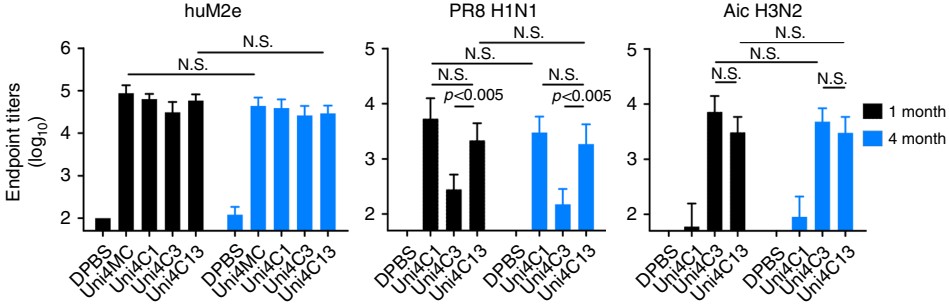

**Fig. 7** Long-lasting immunity. Mice were immunized with different PNps. Antibody endpoints against 4MtG, formalin-inactivated PR8 H1N1 and formalin-inactivated Aic H3N2 were measured in immune sera at time points of 1 and 4 months after immunization. Data are presented as mean ± SD ($n = 10$) and statistical significance was analyzed by $t$-test. N.S. indicates no significance between two compared groups. The experiments were repeated twice with similar results

University Institutional Animal Care and Use Committee (IACUC) under protocol number A16024. Female BALB/c mice (six- to eight-week old) were purchased from Jackson Laboratory and were housed in the animal facility at Georgia State University. Bleeding, infection, and sampling were performed under light anesthesia via inhalation of isoflurane to reduce mouse suffering.

**Design of 4MtG recombinant protein**. Tetrameric tandem M2e polypeptides (4 copies) were stabilized by introducing a foreign tetramerization motif (tetra-) GCN4 at the C-terminal of the recombinant 4MtG protein, as diagrammed in Fig. 1a[58]. The encoding gene of 4MtG was generated by primer extension with overlapping Polymerase chain reaction (PCR). A signal peptide encoding sequence from honeybee melittin was employed to facilitate protein expression and secretion in insect *Spodoptera frugiperda* Sf9 cells (Sf9, ATCC, CRL-1711). A hexa-Histidine tag sequence was added after the GCN4 motif sequence. In each M2e copy of 4MtG, the two site-mutations C17S and C19S were made. The nucleotide sequence of 4MtG is shown in Supplementary Note 1. The four copies of different M2e sequences and their order in 4MtG are listed in Supplementary Table 1 and Fig. 1a. These consensus M2e sequences were made based on 11732 human, 5920 swine, 6267 avian, and 3270 domestic fowl influenza virus M2 sequences deposited in the National Center for Biotechnology Information (NCBI) databank. Molecular evolutionary genetics analysis version 6.0 (MEGA6) was used to align and analyze sequences[59]. 4MtG was purified from recombinant baculovirus (rBV)-based protein expression in insect cells. In brief, the 4MtG encoding sequence was cloned into the transfer vector pFastBac-1 (Invitrogen) for the rBV generation. 4MtG was purified from the rBV-infected insect cell Sf9 culture supernatants by affinity chromatograph.

**Design of head-removed HA recombinant proteins**. According to previous results, trimerization motifs facilitate HA oligomerization in the absence of the HA transmembrane domain[17,18]. A C-terminal sequence containing 6G3S or PGS linker, trimerization motif (tri-) GCN4 and hexa-Histidine tag were added following the hrH1 or hrH3 sequences for oligomerization and purification purposes (diagrammed in Fig. 1b). The coding sequence of the major head domain of H1 HA (GenBank Protein Accession: CAA24272.1, amino acids S53-P321) and the major head domain of H3 HA (GenBank Protein Accession: BAF37221.1, amino acids S61-P322) were replaced with a linker sequence encoding four glycines (4G), which is predicted to be a flexible linker and not disrupt the folding of the remainder of the molecule[18,60]. To inhibit the conformational shift to the post-fusion form, the residues between F61 and L89 in HA2 of H1 and the residues between F63 and I77 in HA2 of H3 were replaced with non-hydrophobic 4G2S and 5G3S linkers, respectively. The F63 and V73 hydrophobic residues in HA2 are largely responsible for stabilizing the coil of the low pH structure[61]. In the neutral pH conformation, these deleted residues are a part of loop B connecting helix A and helix C, and participate in the formation a single long helix ABC in the low pH conformation. The deletion of part of loop B can block the formation of the long helix ABC, fixing the protein structure in the pre-fusion state[61]. The region between R76 and H106 is highly conserved among H3 viruses[62]. Therefore, the loop B truncation in hrH3 ended at the hydrophobic residue I77 to retain the most conserved regions. To further stabilize the hrH3 construct, an intra-disulfide bond was introduced in H3 by site-mutagenesis at V325C in HA1 and S438C in HA2. The encoding gene was generated from the full-length HA genes by overlapping PCR. Recombinant hrH1 and hrH3 were purified from rBV-based insect cell protein expression as done for the 4MtG purification.

**Bis [sulfosuccinimidyl] (BS3) crosslinking**. The oligomeric states of purified 4MtG, hrH1, and hrH3 proteins were determined using the soluble Bis [sulfosuccinimidyl] (BS3) crosslinker (Thermo Scientific, Waltham, MA) in a crosslinking reaction to fix the polymeric structures of proteins followed by reducing sodium dodecyl sulfate polyacrylamide gel electrophoresis (SDS-PAGE)

(Supplementary Fig. 5a, b) and Western blot using anti-His antibody at 1 μg/ml (Cat. No. ab18184, Abcam) (Supplementary Fig. 5c, d), as described previously[32]. Briefly, 1 μg recombinant protein was incubated at room temperature in the presence of 4 mM BS3 for 30 min. The crosslinking reaction was stopped by the addition of 1 M Tris-HCl pH 8.0 to a final concentration of 50 mM.

**PNps fabrication**. PNps were made as previously described with modification[32]. To make the 4MtG core PNps (Uni4MC), the 4MtG protein solution in DPBS (Thermo Scientific, Waltham, MA) was desolvated with a 4:1 volume ratio of absolute ethanol to protein solution. 4.8 ml absolute ethanol was dripped at a constant rate of 1 ml/min into 1.2 ml of 3.2 mg/ml 4MtG protein solution under constant stirring by magnetic stir bar at 600 rpm at room temperature, followed by centrifugation at 15,000 × g for 15 min at room temperature to pellet the PNps. The PNp pellet was resuspended by sonication in either 1 ml 2.8 mg/ml hrH1 in DPBS, 1 ml 3.1 mg/ml hrH3 in DPBS, or 1 ml DPBS to generate hrH1 coated double-layer PNps (Uni4C1), hrH3 coated double-layer PNps (Uni4C3) and uncoated PNps (Uni4MC), respectively. The water-soluble, thiol-cleavable and primary amine-reactive crosslinker 3,3′-dithiobis [sulfosuccinimidylpropionate] (DTSSP; Cat No.21578, Thermo Scientific, Waltham, MA) was used to stabilize the resulting PNps. Crosslinking reactions were performed in 5 mM DTSSP for 1 h while stirring at 4 °C, and were quenched with 30 mM Tris-HCl solution at pH 7.4 for 15 min. Following collection by centrifugation at 20,000 × g for 30 min at 4 °C, PNps were resuspended by sonication in 1 ml DPBS. The fabrication of OVA PNps was described previously[33].

**PNp characterization**. Nanoparticle size distribution was assessed by DLS with a Malvern Zetasizer Nano ZS (Malvern Instruments, Westborough, MA). Protein concentration in the PNp solution was assessed with a BCA assay per the manufacturer's instructions (Thermo Scientific, Waltham, MA). PNps were resuspended in water, air-dried, and sputter-coated with carbon prior to visualization with a Zeiss LEO 1450vp scanning electron microscope (Carl Zeiss, Jena, Germany) at 5.0 kV. Uni4C1, Uni4C3, and Uni4MC PNps were labelled with 1 μg/ml mouse monoclonal antibody C179 (Cat No. M145, TaKaRa), 2 μg/ml 12D1 and the antibody mixture of C179 and 12D1, respectively, and were then labelled with the EM grade gold-conjugated goat-anti-mouse secondary antibody (15 nm, Cat No. 25133, AURION). Ten microliter droplets of stained PNp suspension were adsorbed on a carbon/formvar coated 300-mesh copper grid for 5 min, followed by removal of the remaining liquid with filter paper. Transmission electron microscope LEO 906E (Carl Zeiss) was employed for visualization.

Antigen content of Uni4MC was analyzed in Western blot using anti-M2e antibody at 1 μg/ml (14C2, Cat. No. MA1082, Invitrogen) (Supplementary Fig. 5e), while Uni4C1 and Uni4C3 were analyzed in 10% SDS-PAGE (Supplementary Fig. 5f).

**Pull-down assay**. The pull-down experiment was performed with Dynabeads Protein G (Cat No. 10003D, Thermo Fisher Scientific) according to the manufacturer's instructions. Briefly, 50 μl of Dynabeads protein G suspension was incubated with rotation with 10 μg of C179, 12D1 or a mixture of these two antibodies in 200 μl DPBS with 0.02% Tween-20 for 10 min at room temperature. Tween-20 was used to avoid aggregation. The tube was placed on a magnet and then the supernatant was removed. After removal of the tube from the magnet, 200 μl DPBS with 0.02% Tween-20 was added to resuspend the beads-Ab complex. To avoid co-elution of antibody, 1 mM BS3 crosslinker reagent was added into the beads-Ab complex and incubated for 1 h at room temperature. After the supernatant was removed and collected (Pre-wash sample), 5 μg PNps Uni4C1 or Uni4C3 were added to separate tubes containing beads-Ab complex and incubated with rotation for 1 h at room temperature to allow PNps to bind to the beads-Ab complex. When the tube was placed on the magnet, the supernatant was transferred to a clean tube for further analysis (Unbound fraction sample). The beads-

Ab-PNp complex was washed three times using 200 μl DPBS for each wash, then was resuspended in 100 μl DPBS and transferred into a clean tube. After removal of the supernatant, 20 μl elution buffer (50 mM Glycine at pH 2.8) was added and the beads-Ab-PNp complex was gently resuspended and incubated with rotation for 2 min at room temperature to dissociate the complex. The tube was placed on a magnet and the supernatant containing eluted PNp was transferred to a clean tube (Eluent sample). The collected Pre-wash, unbound fraction and eluent samples were immediately analyzed using Western blot, where 8 μg/ml rabbit anti-His polyclonal antibody (Cat No. PA1-983B, Thermo Fisher Scientific) was used for blotting (Supplementary Fig. 5g).

**Enzyme-linked immunosorbent assay**. The binding of hrHA to HA stem-specific monoclonal antibodies C179 (Cat No. M145, Clontech Labratories, CA), 12D1 and 9H10 (12D1 and 9H10 monoclonal antibodies are kind gifts from Dr. Peter Palese, Icahn school of medicine at Mount Sinai, New York) was tested using a sandwich-ELISA method described previously[19]. C179, 12D1, and 9H10 were used as coating antibodies. In 12D1 or 9H10 coated plates, 50 μl of 1:3 serially diluted FL H3 ectodomain protein, hrH3 and 4MtG were added for binding. In C179 coated plates, diluted inactivated PR8 H1N1, hrH1, and 4MtG were added for binding. A monoclonal anti-His-tag antibody conjugated with horseradish peroxidase (HRP) (Cat No. R931-25, Thermo Scientific, Waltham, MA) was used as the detection antibody. The binding of 4MtG to M2e-specific monoclonal antibody 14C2 (Invitrogen) was tested using a sandwich-ELISA method. Monoclonal antibody 14C2 was used as the coating antibody.

Antibody titers were analyzed using Enzyme-linked immunosorbent assay (ELISA) as described previously[36]. M2e specific antibodies were titrated in immunoplates coated with diverse M2e peptides from human consensus M2e (huM2e), p09, Vtn, and SH. All M2e peptides were synthesized by Synpeptide Co Ltd., Shanghai. PR8 H1, Aic H3, Vtn H5, and SH H7 specific antibodies were titrated in immunoplates coated with different formalin-inactivated viruses, including PR8 H1N1, Aic H3N2, rVn H5N1, and rSH H7N9. The IgG isotype titers were determined by incubating with horseradish peroxidase (HRP)-conjugated goat anti-mouse IgG1 (Cat No. 1071-05, SouthernBiotech, Birmingham, AL) or IgG2a secondary antibodies (Cat No. 1081-05, SouthernBiotech, Birmingham, AL). Antibodies cross-reactive to J57 H2 or GD H10 were titrated in cell-based ELISA. HEK293T cells (ATCC No. CRL-1573) were transfected with engineered pCMV3 plasmid encoding FL H2 (strain: A/Japan/305/1957; Cat No. VG11088-UT, Sino Biological. Inc.) or FL H10 (strain: A/duck/Guangdong/E1/2012; Cat No. VG40351-UT, Sino Biological. Inc.) in Lipofectamine 2000 (Cat No. 11668019, Invitrogen). The transfected HEK293T cells were seeded at a density of $5 \times 10^4$ per well in 96-well plate and was fixed by 80% acetone for 10 min at room temperature prior to serum inoculation.

**Immunization and influenza A virus challenges**. Mice (BALB/c strain, female, 6–8-week-old) received intramuscular (i.m.) immunizations twice, at a 4-week interval, in the hind leg with 50 μl of vaccine mixture in DPBS containing 10.5 μg Uni4MC, 12 μg Uni4C1, 12 μg Uni4C3, 12 μg Uni4C13 (a formulation comprising a mixture of 6 μg Uni4C1 and 6 μg Uni4C3 in DPBS) or 12 μg OVA PNps. Fifty microliter DPBS was used as a placebo. Blood samples were collected at 1 day prior to priming, 3 weeks after priming and boosting and 4 months after boosting to examine the effects on long-term immunity. Four weeks after the boosting immunization, mice were challenged intranasally with $6 \times mLD_{50}$ of mouse adapted (m.a.) influenza A virus strains in 50 μl DPBS. The m.a. strains used for these challenges were PR8[23], p09[34], Aic[63], A/Philippines/2/1982 (Phi, H3N2)[34], rVn (rVtn; HA and neuraminidase (NA) genes were derived from Vtn, and the remaining backbone genes from PR8) or rSH (rSH; HA and NA genes were derived SH, and the remaining backbone genes from PR8).

The rVn H5N1 and rSH H7N9 reassortant viruses were generated and rescued as previously described[64,65]. The backbone plasmid system for generating reassortant virus was based on PR8 virus and generously provided by Dr. Robert Gordon Webster. The H5 HA and N1 NA genes with non-coding regions derived from H5N1 (A/Vietnam/1203/2004) were chemically synthesized and cloned into the pHW plasmids[64]. Plasmids pDZ with genes encoding H7 HA and N9 NA derived from A/Shanghai/2/2013 were previously described[65] and kindly provided by Dr. Adolfo García-Sastre. The eight plasmids with 6:2 reassortant viral gene segments were co-transfected into HEK293T cells using Lipofectamine 2000 (Invitrogen). At day three post transfection, the virus-containing cell culture supernatants were inoculated into the allantoic cavities of ten-day-old embryonated chicken eggs. The rescue of reassortant viruses was determined by hemagglutination of chicken red blood cells. Virus titers were determined by plaque-forming units on Madin-Darby Canine Kidney (MDCK) cells (Cat. No. PAT-6500, ATCC).

Body weight loss and survival rates were monitored daily for 14 days post infection. Weight loss of ≥20% was used as the endpoint at which mice were euthanized per IACUC guidelines.

**Interferon gamma (IFNγ) ELISpot procedures**. The number of IFNγ secreting cells after restimulation was evaluated using an ELISpot method described previously[34]. Briefly, 3 weeks after boosting, splenocytes were isolated from all immunization groups. Each well in a 96-well filtration plate (Catalog Number: MSIPS4W10, Fisher Scientific) was loaded with $5 \times 10^5$ splenocytes for restimulation and a final concentration of 2 μg/ml of an M2e peptide pool (comprised of equal amounts of huM2e, p09, Vtn, and SH M2e peptides), H1 peptide pool (NR-15433 Peptide Array, H1N1 A/California/4/2009, Beiresources, NIAID), H3 peptide pool (NR-19246 Peptide Array, H3N2 A/Brisbane/10/2007, Beiresources, NIAID), or mock-restimulation. The developed plates were rinsed with purified water and air dried before counting using a Bioreader-6000-E (Biosys, Germany).

**Determination of lung virus titers**. Three mice per immunization group were euthanized at day 5 post $1 \times mLD_{50}$ PR8 or Aic infection. Determination of lung virus titers were described previously[29]. The presence of virus in the supernatant was assayed by measuring the hemagglutinating activity in the supernatant, using the Reed and Muench method for calculation[66].

**Histological analysis**. Three mice per immunization group were euthanized at day 5 post $1 \times mLD_{50}$ PR8 or Aic infection. Lung tissues were isolated and fixed with 10% neutral buffered formalin. Fixed lung tissues were embedded in paraffin and processed for Haemotoxylin and Eosin (H&E) staining. Three sections with 10 μm thickness from three different parts of the lungs were stained with H&E and examined microscopically by three unbiased pathologists. The severity of the inflammation in the examined lung sections was scored on a scale of 0 to 5 (with 0.5 interval). Scores were given as absent (0), subtle (1), mild (2), moderate (3), severe (4), and massive (5).

**HAI assay**. HAI titers of mouse immune sera were assayed as previously described[18] with viruses PR8, Aic H3N2, A/Hong Kong/1/1968 (p68, H3N2), rVn or rSH. The lowest serum dilution able to inhibit virus hemagglutination is shown.

**Neutralization assay**. The neutralization assay was described previously[67]. Pooled serum samples were heat-inactivated for 30 min at 56 °C. Mixtures of virus with final concentrations of $100 \times TCID_{50}$ virus per mixture and two-fold serial diluted serum samples (final serum dilution from 1:10 to 1:1280) were incubated for 2 h at 376 °C, 5% $CO_2$ in 50 μl virus medium (DMEM, 100 U/ml penicillin and 100 μg/ml streptomycin), then were subsequently added to the MDCK cells and incubated for 72 h at 376 °C, 5% $CO_2$. A standard hemagglutination assay was performed to measure virus inhibition.

**Antibody-dependent cellular cytotoxicity (ADCC) surrogate assay**. An ADCC surrogate assay was performed according to the kit manufacturer's protocol (Cat No. M1211, Promega) with modification. Briefly, HEK293T cells were maintained in DMEM supplemented with 10% heat inactivated fetal bovine serum (FBS, Invitrogen), 2 mM L-glutamine and 1% (w/v) penicillin/streptomycin stock solution at 376 °C, 5% $CO_2$. Two days before the experiment, HEK293T cells were transfected with plasmid DNA encoding FL H1 (Cat No. VG11684-UT, Sino Biological. Inc.) or H3 proteins (Cat No. VG1707-UT, Sino Biological. Inc.), using Lipofectamine 2000 (Invitrogen) in Opti-MEM (Invitrogen). The M2-expressing MDCK cell[68] culture medium was supplemented with 10 μM Amantadine (Sigma) to support cell growth, 7.5 μg/ml of puromycin (Invitrogen) and 10% FBS at 376 °C, 5% $CO_2$. One day before the assay, transfected HEK293T and M2-expressing MDCK cells were harvested and seeded in sterile white 96-well plates (Costar). After 24 h, serum samples were heat inactivated for 30 min at 566 °C and then serially diluted in assay buffer (4% ultra-low IgG FBS [Promega] in RPMI 1640 [Gibco]). Serum dilutions and a stable Jurkat cell line expressing mouse FcγRIV (Cat No. M1211, Promega) were added and incubated for 6 h at 376 °C at a target-effector ratio of 1:5. Cells were equilibrated to room temperature for 15 min before Bio-Glo Luciferase assay substrate (Promega) was added. Luminescence was read out after 10 min on a GloMax (Promega). Data are expressed as luminescence RLU of signal in the absence of serum.

**Depletion of AM**. Depletion of AM was performed as described previously[56]. Briefly, BALB/c mice were anesthetized by intraperitoneal (i.p.) injection with ketamine/xylazine and then 100 μl PBS-liposomes or clodronate-liposomes were administered slowly intratracheally (i.t.). Twenty-four hours after liposome administration, mice were injected i.p. with 400 μl pre-immune serum or Uni4C13 serum. After 24 h, mice were bled to determine serum titers by using M2e peptide ELISA, and then were challenged with $3 \times mLD_{50}$ Phi H3N2.

**Statistical analysis**. All data plotted with error bars are expressed as means with standard derivation. The P values were generated by analyzing data with a two-tail unpaired t test using the Prism 5 program (GraphPad software). Survival rate statistical analysis was performed with Kaplan-Meier analysis.

**Data availability**. The authors declare that the data supporting the findings of this study are available within the article and its Supplementary Information files, or are available from the authors upon request.

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

## Acknowledgements
We thank Robert Simmons, Ping Jiang and Debby Walthall from Georgia State University, Atlanta, Gene S. Tan and Chen Wang from Icahn school of medicine at Mount Sinai, New York, Kijoon Cho from Mogam Institute, Yongin-si, South Korea, for excellent technical assistance and conceptual advice, Peter Palese from Icahn school of medicine at Mount Sinai, New York for providing mouse monoclonal antibodies 12D1 and 9H10, Adolfo García-Sastre from Icahn school of medicine at Mount Sinai, New York for HA and NA genes of H7N9 (A/Shanghai/2/2013), and Robert Gordon Webster from St. Jude Children's Research Hospital in Memphis, Tennessee, for eight-plasmid reassortant system. This work was supported by the Institute of Biomedical Sciences, Georgia State University and by grants R01AI101047, R01AI116835, and R01AI093772 from US National Institutes of Health/National Institute of Allergy and Infectious Diseases.

## Author contributions
B.Z.W. was responsible for conception and experimental strategy of this study. L.D., T.M., T.Z.C., R.W.C., S.M.K., J.A.C., and B.Z.W. conceived and designed the experiments. L.D., T.M., G.X.G., Y.W., and Y.M.K performed the experiments. L.D., T.M., T.Z.C., G.X.G., Y.W., Y.M.K., R.W.C., and B.Z.W. analyzed the data. L.D., Y.M.K., S.M.K., and B.Z.W. contributed reagents/materials/analysis tools. L.D., G.X.G., S.M.K., R.W.C., and B.Z.W. wrote the manuscript. L.D. made the figures and table.

## Additional information

**Competing interests:** L.D. and B.Z.W. are inventors on patents and patent applications related to this study. All other authors have no potential conflict of interest.

