## [Peer Review File · Nature Communications]

Reviewers' comments:

Reviewer #1 (Remarks to the Author):

In their manuscript Deng et al. describe a novel universal influenza virus vaccine candidate based on double-layered protein nanoparticles with an M2e core and a headless-HA coating. The tested construct induces broad binding and protection in mice and the authors show that this protection might be mostly based on antibody effector functions. The manuscript is well written and the experiments are nicely designed. However, there are several issues that need the author's attention.

Major points

- 1) It seems that the abstract is missing.
- 2) The authors tested the H3 headless HA construct with mAb 12D1. This mAb is NOT conformation dependent. Another mAb (e.g. CR8020, CR8043, 9H10) should be used.
- 3) The authors' constructs possess HA tags and trimerization domains which are immunogenic. The serology to HAs is done using recombinant HAs which likely have HA tag/trimerization domains and a good proportion of the reactivity might be against those. Please a) specify the exact amino acid sequence of the recombinant HAs (so that it is clear where tag-specific reactivity is possible) and b) show binding against preparations of inactivated virus.
- 4) It is unclear if A/Aichi and A/Philippines were used (those are usually not mouse lethal) or if it was X-31 and X-79 which have internal genes (plus M2e) from A/PR/8/34.
- 5) Please indicate clearly that the H5 and H7 challenge viruses expressed the PR8 M2e. In connection to that please clarify if Ex. Data Table 1 shows wild type of PR8 M2e for these viruses.
- 6) While the clodronate depletion experiment is very nice it is unclear if the removal of macrophages makes the mice more susceptible in general. This makes the result questionable. The authors should try to design a better controlled experiment.

Minor points

- 1) Line 1: 'protein', not 'Protein'
- 2) Line 32: What does highly conserved mean in this context? Please specify.
- 3) Line 41: 'A/Puerto Rico/8/1934', not 'A/Puerto Rico/08/1934'
- 4) Line 42: 'A/Aichi/2/1968', not 'A/Aichi/02/1968'

- 5) Figure 1: Which numbering is used for the constructs in A? H3 numbering? Please specify this in the figure legend.
- 6) Line 68 and below: Several abbreviations are not defined including DTSSP, DPBS
- 7) Line 96: 'A/California/07/2009', not 'A/California/07//2009'
- 8) Figure 2: It is explained why H5, H7 and H10 were included. Please explain to the reader why H2 was included.
- 9) Figure 3i: For clarity it would be better to separate the reactivity out by substrate.
- 10) Line 189: Why is the data not shown?
- 11) Line 189: Please use 'low income countries' instead of 'developing countries'
- 12) Ext. Data Fig 1a: Why was a tetramerization domain used? Please clarify.
- 13) Ext. Data Fig 4e: It is unclear why Uni4C1 would have a drastically different IgG1:iG2a ration against H3. This is not the case for Uni4C3 against H1. Are the authors sure this data is correct?

Reviewer #2 (Remarks to the Author):

The manuscript by Deng et al. describes the novel universal influenza vaccines based upon M2e tetramer molecules and stalk based approaches for vaccination. This proof of concept study demonstrates that these vaccines show amelioration of diseases and weight loss against multiple challenges from viruses of different subtypes. Several of these viruses are mouse adapted, except for the as the the pnH1N1 and H5N1 and H7N9 strains. In general there is a correlation with neutralization titers, except for H7N9.

The authors discuss ADCC as a possible explanation to the results and show some data to this effect. The study would be strengthened using a ferret model, rather than mouse, use real human seasonal viruses, examine shedding from the nasal turbs of ferrets, and show lack of transmission. The mouse results demonstrate that this proof of concept approach has merit. more rigorous testing will be needed.

Reviewer #3 (Remarks to the Author):

The manuscript entitled 'Double-layered Protein nanoparticles induce broad protection against divergent influenza A viruses' is interesting and presents a concept for an approach leading to a universal influenza vaccine.

Both M2e and HA have been proposed by the authors and others as targets for universal

vaccination strategies to protect broadly from the disease. It also appears that the nanoparticle-aggregates described here have been previously published in somewhat similar form by the authors.

Wang, L. et al. Coated protein nanoclusters from influenza H7N9 HA are highly immunogenic and induce robust protective immunity. *Nanomedicine : nanotechnology, biology, and medicine* 13, 253-262, doi:10.1016/j.nano.2016.09.001 (2017).

Wang, L. et al. Nanoclusters self-assembled from conformation-stabilized influenza M2e as broadly cross-protective influenza vaccines. *Nanomedicine : nanotechnology, biology, and medicine* 10, 473-482, doi:10.1016/j.nano.2013.08.005 (2014).

The in vivo efficacy results are promising and indicate broad cross reactivity leading to generation of sera broadly reactive toward several tested subtypes. Furthermore prophylaxis potency was demonstrated in challenge studies. While the in vivo efficacy and prophylaxis data demonstrate high potential of the approach, the novelty is not clear.

Figure 3 is extremely difficult to read, because each panel uses its own color scheme; the color scheme/legend should be consistent throughout.

The 'nanoparticle' synthesis and characterization is ambiguous and several points require clarification and/or additional experimentation to support the claims made:

1) In one statement the author claim their protein assembly to resemble a virion – in another statement they claim these crosslinked protein aggregates to be superior to virions. Many virions are characterized by their inherent stability in various environments – because no stability data, neither during storage, nor in biological media is presented, such claims are not supported by any data. Further, a characteristic of virions and their virus-like particles is their symmetry and well-defined particulate structures. Based on the nature of the crosslinked protein aggregate a comparison to viral structure is not supported.

2) Further characterization of the structural properties of the protein aggregates is needed: The molecular composition of the Uni4MC1/3 protein aggregates is ill defined: how many 4MtG per Uni4MC? How many hrH1/3 per coated PNP? In particular the molecular composition of the double layer assembly needs to be characterized in more detail: the SEM shows a large distribution of sizes; couldn't it be possible that a mix of particles is produced: some double-layer particles, some free or intercrosslinked Uni4MC particles, and some free or intercrosslinked hrHA complexes? The question arises in particular because homo-bi-functional linkers are used to crosslink the particles; given there is no control over directionality, each protein could be crosslinked with itself rather than being coupled to the other species. The SDS gel shows both proteins are present yet does not provide any evidence that a double-layer was formed.

Possible suggestions would be: i) a pull down assay could be done, where hrH1/3 is pulled down (e.g. using a capture antibody), then the pulled down samples are analyzed and amount of hrH1/3 vs Uni4MC is quantified and compared to the amounts of both proteins found free in solution. ii) Immunogold labeling and TEM imaging may be a possibility to demonstrate that both components are present; although it will be challenging to provide quantitative data. iii) Another possible assay may be to label each protein with a distinct

fluorophore and perform size exclusion chromatography and show that all elements co-elute?

There may be other possible assays – nevertheless additional data should be presented to support that the double layers of proteins are truly formed, to which degree, and to what level of homogeneity?

3) Size distribution: DLS and SEM are presented but do not appear to be in agreement? Most aggregates shown in SEM appear smaller than what is shown in DLS. Quantitative data should be prepared from SEM image analysis and compared to DLS. Of course one would expect the DLS data to show slightly larger particles due to the hydration layer, nevertheless there should be correlation between the data. SEM images of the various Uni4MC/1/3 complexes should be shown at the same magnification.

4) The abbreviations need to be defined clearly in the Figure 1 or its legend; e.g. aggregated 4MtG = Uni4MC PNp; Figure 1e might be mislabeled; what is 4MtG PNp - should this read Uni4MC PNp? Uni4MC vs Uni4C1 and Uni4C3 is not defined (it is defined later in the text, but should also be defined in the Figure).

5) Tetramer and trimer formation. The SDS PAGE indeed indicates that the majority of 4MtG is a tetramer; yet monomer, dimer, and trimer exist. These should be quantified. Even better the tetramer should be purified for use as a 4MtG particle.

For the trimer: it would be better to run this out on a gel with pore sizes suitable for larger proteins. Crosslinking of hrH1 and hrH3 eliminates the presence of the monomer. For hrH1 a dimer is present at ~ 75 kDa; then there is two additional bands - one just below 100 and one higher than 100 kDa. The question arises whether these are trimers and multimers? Further investigation is required to confirm the presence of trimers. Same applies to the hrH3 particle.

6) The native antigenicity was confirmed for the hrH1/3 but not 4MtG - this should be included.

Responses to reviewer #1's comments:

Reviewer #1 (Remarks to the Author):

In their manuscript Deng et al. describe a novel universal influenza virus vaccine candidate based on double-layered protein nanoparticles with an M2e core and a headless-HA coating. The tested construct induces broad binding and protection in mice and the authors show that this protection might be mostly based on antibody effector functions. The manuscript is well written and the experiments are nicely designed. However, there are several issues that need the author's attention.

Major points

Question: 1) It seems that the abstract is missing.

Author Response: The Abstract section has been separated from the main article.

Question: 2) The authors tested the H3 headless HA construct with mAb 12D1. This mAb is NOT conformation dependent. Another mAb (e.g CR8020, CR8043, 9H10) should be used.

Author Response: The binding ability of hrH3 to a conformation dependent mAb 9H10 was tested in sandwich ELISA. The result has been added in Extended Data Figure 2g. Corresponding corrections have been made at lines 52, 331, 333, 334, 540 and 583 in the corrected manuscript.

Question: 3) The authors constructs possess his tags and trimerization domains which are immunogenic. The serology to HAs is done using recombinant HAs which likely have his tag/trimerization domains and a good proportion of the reactivity might be against those. Please a) specify the exact amino acid sequence of the recombinant HAs (so that it is clear where tag-specific reactivity is possible) and b) show binding against preparations of inactivated virus.

Author Response: We agreed with the reviewer. In the revised version, inactivated viruses or transfected cell-expressed HA were utilized as capture antigens in ELISA or cell surface ELISA (Figure 2a, b and Extended data figures 4 and 8 in the revised version). The experiment method was described in page 17 line 344 to page 18 line 351. Other corresponding corrections have been made in text at lines 118, 121 and 126, and in the legend of Extended data Fig 4.

These recombinant HA antigens were not used in the antibody measurement in this version thus are not described.

Question: 4) It is unclear if A/Aichi and A/Philippines were used (those are usually not mouse lethal) or if it was X-31 and X-79 which have internal genes (plus M2e) from A/PR/8/34.

Author Response: Both viruses are mouse adapted strains (described in page 8 line 143, and page 18 lines 361 to 365 in the new version).

Question: 5) Please indicate clearly that the H5 and H7 challenge viruses expressed the PR8 M2e. In connection to that please clarify if Ex. Data Table 1 shows wild type of PR8 M2e for these viruses.

Author Response: This has been addressed as suggested (Page 8 line 152 to 153 in the revised manuscript).

Question: 6) While the clodronate depletion experiment is very nice it is unclear if the removal of macrophages makes the mice more susceptible in general. This makes the result questionable. The authors should try to design a better controlled experiment.

Author Response: Our results indicated that, after lethal infection of influenza A virus Phi H3N2, alveolar macrophage-depleted mice (Clodronate/preimmune group) did not show significantly severer weight loss than control mice (DPBS/preimmune group), though some control mice died one day slower. Therefore, the effects of alveolar macrophage removal on mouse susceptibility to lethal dose Phi H3N2 infection is subtle. Passive transfer of Uni4C13 immune sera protected DPBS- but not Clodronate-treated mice, indicating that the necessity of alveolar macrophages for immune serum-mediated protection. At this point, mouse groups of Clodronate-treatment without immune serum transfer and DPBS-treatment with or without immune serum transfer are appropriate controls.

Minor points

Question: 1) Line 1: 'protein', not 'Protein'

Author Response: This has been corrected as suggested (line 1).

Question: 2) Line 32: What does highly conserved mean in this context? Please specify.

Author Response: It means that the M2e sequences from all human seasonal influenza A viruses are conserved with very few amino acid residue differences. Corresponding clarification has been made at lines 30 and 31 in corrected manuscript.

Question: 3) Line 41: 'A/Puerto Rico/8/1934', not 'A/Puerto Rico/08/1934'

Author Response: This has been corrected (line 41 in revised manuscript).

Question: 4) Line 42: 'A/Aichi/2/1968', not 'A/Aichi/02/1968'

Author Response: This mistake has been corrected at line 42 in the new version.

Question: 5) Figure 1: Which numbering is used for the constructs in A? H3 numbering? Please specify this in the figure legend.

Author Response: The numbering explanation is added in the legend of Fig 1a in the corrected manuscript.

Question: 6) Line 68 and below: Several abbreviations are not defined including DTSSP, DPBS

Author Response: They have been defined at their first use in the new version (page 4 line 73 and page 8 line 148, respectively).

Question: 7) Line 96: 'A/California/07/2009', not 'A/California/07//2009'

Author Response: This typing error is corrected (line 113 in the corrected manuscript).

Question: 8) Figure 2: It is explained why H5, H7 and H10 were included. Please explain to the reader why H2 was included.

Author Response: The explanation for including H2 has been added in the legend of Fig. 2 at lines 130 to 133 in the corrected manuscript.

Question: 9) Figure 3i: For clarity it would be better to separate the reactivity out by substrate.

Author Response: The ADCC reactivity has been separated by substrate in Figure 3i, j, k. The corrections have been made at page 10 lines 171 to 173 in the legend and at page 10 lines 185 and 186 in article text.

Question: 10) Line 189: Why is the data not shown?

Author Response: A separate manuscript is in preparation to study the stability of protein nanoparticles. The protein antigens in nanoparticles in that work are not exactly the same as used in the current manuscript but are comparable when discussing the stability of protein nanoparticles. The 'data not shown' has been deleted in the revised manuscript. Since the stability data is not publicly available yet, we have included it as supplemental information for reviewers' reference.

Question: 11) Line 189: Please use 'low income countries' instead of 'developing countries'

Author Response: This has changes as suggested at page 12 line215.

Question: 12) Ext. Data Fig 1a: Why was a tetramerization domain used? Please clarify.

Author Response: This typing error has been corrected in Ext. Data Fig 1a.

Question: 13) Ext. Data Fig 4e: It is unclear why Uni4C1 would have a drastically different IgG1:iG2a ration against H3. This is not the case for Uni4C3 against H1. Are the authors sure this data is correct?

Author Response: Based on the inactivated virus-coated ELISA results, the antibody IgG1 and IgG2a titers of Uni4C1 serum against H3 and Uni4C3 serum against H1 show no significance (Extended data Fig. 4 d and 4e in the revised manuscript).

Response to reviewer #2's comments:

Reviewer #2 (Remarks to the Author):

The manuscript by Deng et al. describes the novel universal influenza vaccines based upon M2e tetramer molecules and stalk based approaches for vaccination. This proof of concept study demonstrates that these vaccines show amelioration of diseases and weight loss against multiple challenges from viruses of different subtypes. Several of these viruses are mouse adapted, except for the as the the pnH1N1 and H5N1 and H7N9 strains. In general there is a correlation with neutralization titers, except for H7N9.

The authors discuss ADCC as a possible explanation to the results and show some data to this effect. The study would be strengthened using a ferret model, rather than mouse, use real human seasonal viruses, examine shedding from the nasal turbs of ferrets, and show lack of transmission. The mouse results demonstrate that this proof of concept approach has merit. more rigorous testing will be needed.

Author Response: All six types of influenza A virus used in the challenge experiments are mouse-adapted, which has been described in lines 361 to 365 in the Methods section and also at lines 143 and 144 in the manuscript text. We agreed that not only neutralizing immunity but also non-neutralizing immunity play important roles in protection against influenza A virus infection. With our promising *in vivo* data obtained in the mouse model, next we will test this vaccine approach in other laboratory animal models such as ferrets to evaluate the PNp vaccine broad protection efficacy, viral transmission inhibition, and investigate the mechanisms of protection.

Responses to reviewer #3's comments:

Reviewer #3 (Remarks to the Author):

The manuscript entitled 'Double-layered Protein nanoparticles induce broad protection against divergent influenza A viruses' is interesting and presents a concept for an approach leading to a universal influenza vaccine.

Both M2e and HA have been proposed by the authors and others as targets for universal vaccination strategies to protect broadly from the disease. It also appears that the nanoparticle-aggregates described here have been previously published in somewhat similar form by the authors.

Wang, L. et al. Coated protein nanoclusters from influenza H7N9 HA are highly immunogenic and induce robust protective immunity. *Nanomedicine : nanotechnology, biology, and medicine* 13, 253-262, doi:10.1016/j.nano.2016.09.001 (2017).

Wang, L. et al. Nanoclusters self-assembled from conformation-stabilized influenza M2e as broadly cross-protective influenza vaccines. *Nanomedicine : nanotechnology, biology, and medicine* 10, 473-482, doi:10.1016/j.nano.2013.08.005 (2014).

The *in vivo* efficacy results are promising and indicate broad cross reactivity leading to

generation of sera broadly reactive toward several tested subtypes. Furthermore prophylaxis potency was demonstrated in challenge studies. While the in vivo efficacy and prophylaxis data demonstrate high potential of the approach, the novelty is not clear.

Author Response: We studied the immunogenicity of M2e desolvated PNps (Wang et al, 2014) and coated HA as an outer layer to HA desolvated PNps as an improved vaccine approach (Wang et. al, 2017). In this study, it is the first time to combine conserved structure-stabilized HA stalk domains with M2e in two-layered protein nanoparticles as a universal influenza A vaccine approach. Because of the advantage of disassemblable protein nanoparticles as vaccines or protein drug delivery platform and the broad protection observed as described in the revised manuscript, we deeply believe that the data will benefit to the scientific community of vaccine development.

Question: Figure 3 is extremely difficult to read, because each panel uses its own color scheme; the color scheme/legend should be consistent throughout.

Author Response: As suggested, the colors have been changed to be consistent in different panels (Figure 3 (a-f) and Extended Data Figure 5 (a-f) in the revised version).

Question: The 'nanoparticle' synthesis and characterization is ambiguous and several points require clarification and/or additional experimentation to support the claims made:

1) In one statement the author claim their protein assembly to resemble a virion – in another statement they claim these crosslinked protein aggregates to be superior to virions. Many virions are characterized by their inherent stability in various environments – because no stability data, neither during storage, nor in biological media is presented, such claims are not supported by any data. Further, a characteristic of virions and their virus-like particles is their symmetry and well-defined particulate structures. Based on the nature of the crosslinked protein aggregate a comparison to viral structure is not supported.

Author Response: We agreed that the comparison of a random crosslinked protein nanoparticle to a highly organized virion in their structure is not appropriate although their size and multivalent display of surface protein molecules to the immune system have some similarities. We have clarified that the comparison is limited to their size and surface antigenic protein display (page 4 line 79).

Question: 2) Further characterization of the structural properties of the protein aggregates is needed: The molecular composition of the Uni4MC1/3 protein aggregates is ill defined: how many 4MtG per Uni4MC? How many hrH1/3 per coated PNp? In particular the molecular composition of the double layer assembly needs to be characterized in more detail: the SEM shows a large distribution of sizes; couldn't it be possible that a mix of particles is produced: some double-layer particles, some free or intercrosslinked Uni4MC particles, and some free or intercrosslinked hrHA complexes? The question arises in particular because homo-bi-functional linkers are used to crosslink the particles; given there is no control over directionality, each protein could be crosslinked with itself rather than being coupled to the other species. The SDS gel shows both proteins are present yet does not provide any evidence that a double-layer was formed.

Possible suggestions would be: i) a pull down assay could be done, where hrH1/3 is pulled down (e.g. using a capture antibody), then the pulled down samples are analyzed and amount of hrH1/3 vs Uni4MC is quantified and compared to the amounts of both proteins found free in solution. ii) Immunogold labeling and TEM imaging may be a possibility to demonstrate that both components are present; although it will be challenging to provide quantitative data. iii) Another possible assay may be to label each protein with a distinct fluorophore and perform size exclusion chromatography and show that all elements co-elute?

There may be other possible assays – nevertheless additional data should be presented to support that the double layers of proteins are truly formed, to which degree, and to what level of homogeneity?

Author Responses:

As the reviewer suggested, we did more experiments and analysis to further characterize the composition and structural characteristics of the protein nanoparticles. We did immuno-gold labelling and transmission electronic microscopy images, showing the coating of hrH1 or hrH3 onto the surfaces of Uni4MC core PNPs (Extended Data Figure 3i in the new version). The figure legend has been updated accordingly. The experiment procedure has been added in the section 'PNP characterization' in Method (page 15 line 299 to page 16 line 305).

HrH1 or hrH3 could crosslink into oligomeric aggregates in crosslinking reaction, but these oligomers cannot be pelleted down without a desolvated PNP core due to high hydration, which was described in a previous publication (Deng L., et al., 2017 Virology). The explanation has been added into the revised version (page 5 lines 103 and 104). We also added the pull-down assay and Western Blot analysis as suggested. These results demonstrate that the majority of PNPs is not inter-crosslinked, and there is no uncoated 4MtG PNPs in the Uni4C1 and Uni4C3 preparations (Figure 1e, Extended data Figure3h).

Question: 3) Size distribution: DLS and SEM are presented but do not appear to be in agreement? Most aggregates shown in SEM appear smaller than what is shown in DLS. Quantitative data should be prepared from SEM image analysis and compared to DLS. Of course one would expect the DLS data to show slightly larger particles due to the hydration layer, nevertheless there should be correlation between the data. SEM images of the various Uni4MC/1/3 complexes should be shown at the same magnification.

Author Response: The sizes of Uni4MC, Uni4C1 and Uni4C3 in SEM images were quantified using ImageQuantTL software (Extended Data Figure 3f), and were found no significant difference among these groups (Extended Data Figure 3g). These results have been discussed in the revised manuscript at page 5 lines 93 to 98.

As suggested, the bar scales in different SEM images are shown at the same magnification in the revised version (Figure 1e).

Question: 4) The abbreviations need to be defined clearly in the Figure 1 or its legend; e.g. aggregated 4MtG = Uni4MC PNP; Figure 1e might be mislabeled; what is 4MtG PNP - should

this read Uni4MC PNP? Uni4MC vs Uni4C1 and Uni4C3 is not defined (it is defined later in the text, but should also be defined in the Figure).

Author Response: The Uni4MC is defined at its first appearance at line 61 in the corrected manuscript. The label name of lane 3 in Figure 1f has been changed to Uni4MC. The Uni4C1 and Uni4C3 are defined in this version (page 4 lines 66 and 67).

Question: 5) Tetramer and trimer formation. The SDS PAGE indeed indicates that the majority of 4MtG is a tetramer; yet monomer, dimer, and trimer exist. These should be quantified. Even better the tetramer should be purified for use as a 4MtG particle.

For the trimer: it would be better to run this out on a gel with pore sizes suitable for larger proteins. Crosslinking of hrH1 and hrH3 eliminates the presence of the monomer. For hrH1 a dimer is present at ~ 75 kDa; then there is two additional bands - one just below 100 and one higher than 100 kDa. The question arises whether these are trimers and multimers? Further investigation is required to confirm the presence of trimers. Same applies to the hrH3 particle.

Author Response: The oligomer percentages were analyzed using GelQuant software, 89.3% for tetramer, 3.4% for trimer, 3% for dimer and 4.3% for monomer. Although the tetramers dominated the purified protein, we agreed with the constructive advice to purify tetramers as 4MtG protein for future use.

The main thick bands located at above 100 kDa in the crosslinked hrH1 and hrH3 samples are the trimeric hrHA. We supplemented Western Blot analysis of hrHA and crosslinked hrHA (Extended Data Figure 2c). A faint band between 50 and 75 kDa is the dimer, and the thick band above 100 kDa is the trimer. The legend of Extended data figure 2c has been updated, and related correction was made at page 3 line 48.

Question: 6) The native antigenicity was confirmed for the hrH1/3 but not 4MtG - this should be included.

Author Response: The antigenicity of 4MtG was confirmed in M2e specific monoclonal antibody 14C1 coated sandwich ELISA. The result was added in Extended Data Figure 2d in the revised version. The legend of Extended data figure 2 has been updated accordingly.

REVIEWERS' COMMENTS:

Reviewer #1 (Remarks to the Author):

The authors addressed all reviewers' comments.

Reviewer #3 (Remarks to the Author):

The authors have sufficiently addressed all previous concerns. The manuscript is technically sound. Given the prior work by the authors the work presents a natural extension of their previous efforts.